# Exploring the acute affective responses to resistance training: A comparison of the predetermined and the estimated repetitions to failure approaches

Hadar Schwartz [1,2], Aviv Emanuel [1,2,3], Isaac Isur Rozen Samukas[1,2], Israel Halperin [1,2]*

1 School of Public Health, Sackler Faculty of Medicine, Tel-Aviv University, Tel-Aviv, Israel, 2 Sylvan Adams Sports Institute, Tel Aviv University, Tel-Aviv, Israel, 3 School of Psychological Sciences, Tel-Aviv University, Tel-Aviv, Israel

* ihalperin@tauex.tau.ac.il

## Abstract

### Background

In resistance-training (RT), the number of repetitions is traditionally prescribed using a predetermined approach (e.g., three sets of 10 repetitions). An emerging alternative is the estimated repetitions to failure (ERF) approach (e.g., terminating sets two repetitions from failure). Despite the importance of affective responses experienced during RT, a comparison between the two approaches on such outcomes is lacking.

### Methods

Twenty women (age range: 23–45 years) without RT experience completed estimated one repetition maximum (RM) tests in four exercises. In the next two counterbalanced sessions, participants performed the exercises using 70%1RM. Participants completed ten repetitions in all three sets (predetermined condition) or terminated the sets when perceived to be two repetitions away from task-failure (ERF condition). Primary outcomes were affective-valence, enjoyment, and approach-preference and secondary outcomes were repetition-numbers completed in each exercise.

### Results

We observed trivial differences in the subjective measures and an approximately even approach-preference split. Under the ERF condition, we observed greater variability in repetition-numbers between participants and across exercises. Specifically, the mean number of repetitions was slightly lower in the chest-press, knee-extension, and lat-pulldown (~1 repetition) but considerably higher in the leg-press (17 vs. 10, $p<0.01$).

**Data Availability Statement:** All relevant data are within the paper and its Supporting information files.

**Funding:** This study was supported by grants from the Israeli Science Foundation (1249/20) and the Marguerite Stolz Research Fund, Sackler Faculty of Medicine, Tel Aviv University. The funders had no role in study design, data collection and analysis, decision to publish, or preparation of the manuscript.

**Competing interests:** The authors have declared that no competing interests exist.

## Conclusions

Both approaches led to comparable affective responses and to an approximately even approach preference. Hence, prior to prescribing either approach, coaches should consider trainee's preferences. Moreover, under the ERF condition participants completed a dissimilar number of repetitions across exercises while presumably reaching a similar proximity to task-failure. This finding suggests that ERF allows for better effort regulation between exercises.

## Introduction

The number of repetitions to complete per set and exercise is one of the key variables in designing and prescribing resistance-training (RT) programs. Professional organizations, such as the American College of Sports Medicine, advocate the prescription of a fixed and predetermined number of repetitions before session or set initiation (e.g., three sets of 10 repetitions) [1, 2]. In an attempt to personalize the loads to be lifted, trainees are instructed to use a certain percentage of their predetermined or predicted one repetition maximum (1RM), which is the heaviest load they can lift once (e.g., 70% of 1RM) [1, 2]. However, studies report considerable variability in the number of repetitions trainees can complete to task failure (TF) even when using the same percentage of 1RM [3, 4]. Note that here we refer to TF as an umbrella term that includes not being able to complete another repetition despite attempting to (also known as "momentary failure"), or not attempting the next repetition assuming it could not be completed (also known as "repetition maximum") [5]. Considering the variability in repetitions to TF, using a predetermined number could lead to different proximity to TF [3, 4] resulting in different perceived effort [6] and possibly dissimilar training outcomes.

To illustrate, consider two trainees instructed to complete ten repetitions in a given exercise using 70% of 1RM. One trainee can complete 20 repetitions to TF, so terminating the set after ten repetitions will correspond to a reserve of ten repetitions before reaching TF. Conversely, the other trainee can only complete eight repetitions, so the predetermined goal could not be achieved, and the set will terminate at TF. Terminating sets at different proximities to TF requires different levels of actual effort (i.e., terminating a set further away from TF requires relatively less effort compared to a set terminated at a closer proximity to TF). Reaching different levels of actual effort in RT influences trainees' ratings of perceived exertion (RPE) [6], affective [7–9] and physiological [10–12] responses. Those differences could alter exercise-adaptations [13] and psychological outcomes such as perceptions of autonomy and competence [14]. For example, trainees might feel unchallenged or bored in case they are prevented from fulfilling the repetitions potential of the set (i.e., by a fixed number of repetitions) [15], or stressed and incompetent in the case they are pushed to premature TF (i.e., if they cannot complete the fixed number of repetitions due to natural variability in abilities) [14, 16]. These responses, in turn, may also influence the likelihood of adherence to RT programs [17]. Therefore, there is a need to explore repetition prescription strategies that can better account for individual differences in affective responses to RT.

An emerging alternative to the predetermined approach is to prescribe the number of repetitions relative to TF (e.g., terminating a set two or three repetitions away from TF). This approach has been implemented using the Estimated Repetitions to Failure (ERF) [18], and the Repetitions in Reserve (RIR) scales [19] which share similarities. Here, we will use the term

ERF as it better represents the methodology we employed. ERF holds the potential to better regulate intensity of effort [18, 19] and possibly the affective responses it elicits. First, using the ERF approach can better account for individual abilities as there is no restriction upon the number of repetitions to be completed as long as one reaches the specific proximity to TF. Second, the ERF approach might lead to more positive experiences in RT sessions as it may elicit a greater sense of control over one's actions (i.e., autonomy) [e.g., 16, 17]. Allowing people to control their actions by providing them with certain choices regarding their surrounding increases psychological well-being [14, 20, 21] and positive affective responses [22]. Since trainees decide when to terminate a set based on their perceived distance from TF, it can be viewed as an autonomy-supportive process. The latter is of great importance as positive affect experienced in a range of activities, including RT, is correlated with future intention and adherence to exercise [17, 22, 23]. Given that only 30% or less of the world's population are meeting the general RT recommendations [24–26], exploring how ERF influences affective responses is also of public health value.

A growing number of studies [27–31], including a recent meta-analysis [32] have examined trainees' ERF predication accuracy, as well as the long term effects of following ERF on strength and power production among different populations [33–36]. However, despite its potential to influence various psychological outcomes, excluding a few examples [34, 37], the topic remains relatively underexplored. In view of the limited research on this topic, the primary aim of this study was to explore the effects of the ERF, and the predetermined RT prescription approaches, on acute affective responses measured during and after RT sessions, among a cohort of women inexperienced in RT. The secondary aim was to compare the number of repetitions completed under the two conditions in different exercises. To achieve these goals, we implemented RT sessions that are representative of how both approaches are commonly used in practice (i.e., exercise selection [34, 38], loads [2, 7, 22], predetermined repetition numbers and sets [2, 39]). We estimated that under the ERF condition the affective responses will be more positive and that the number of repetitions will vary to a greater extent within participants, as opposed to the predetermined condition.

## Materials and methods

### Study design

This was a randomized, counterbalanced, within-subject, cross-over design. Participants first attended a 1RM prediction session, followed by two experimental conditions. In the predetermined condition, the number of repetitions was fixed to ten for all sets in four exercises: leg-press, knee-extension, chest-press and lat-pulldown. In the ERF condition, participants were instructed to terminate the set when they estimated to be two repetitions away from TF. This value was selected as it was expected to be demanding enough to be aligned with common RT recommendations [1, 19], yet not overly demanding leading to negative affect or physiological experiences (e.g., fatigue or discomfort) [40].

### Participants

A sample of 22 healthy women with extensive Pilates experience but without RT experience volunteered to participate in this study of which 20 completed all three sessions (Table 1). We selected this sample since we were interested in the responses of women, who are often underrepresented in RT studies [41]. Moreover, since inexperienced trainees are less adherent to RT, investigating this population segment is of added value [42]. We decided upon 22 participants as we were aware of our recruitment abilities and resources [43]. Nevertheless, this sample size is common in exercise science studies in which discoveries of non-null effects often

**Table 1. General demographics.**

| | |
|---|---|
| Age | 34.4±6.5 (23–45) |
| Height (cm) | 162.0±5.6 (151–172) |
| Weight (kg) | 58.6±9.0 (44–74) |
| BMI | 22.5±2.8 (19–29) |
| Weekly training sessions (non-RT) | 3.1±0.9 (2–4) |
| 5RM Leg Press (kg) | 83.1±19.9 (50–120) |
| Predicted 1RM (kg) | 93.5 ±22.4 (56–135) |
| 5RM Knee Extension (kg) | 51.1±10.6 (32–72) |
| Predicted 1RM (kg) | 57.5±12.0 (36–82) |
| 5RM Chest Press (kg) | 32.0±9.8 (20–52) |
| Predicted 1RM (kg) | 36.0±11.0 (22–60) |
| 5RM Lat Pull Down (kg) | 27.1±8.5 (13–40) |
| Predicted 1RM (kg) | 30.5±9.6 (14–45) |

Female participants (N = 20). Values are presented as mean±SD (range)

occur. Inclusion criteria were age 18–45, no former orthopedic injuries and no former experience in RT. We excluded pregnant women, injured women and those who reported any RT experience. Participants' background in physical activity consisted of Pilates, aerobic exercise, and dance. All participants were informed of the benefits and the risks of the investigation prior to signing the informed consent form on the first session. Two participants dropped out after the first session due to physical inconvenience experienced during the session. This study was approved by Tel Aviv University Ethics Committee (number 0001540–1).

## Procedures

Participants completed three sessions (a single 1RM prediction session and two experimental conditions) with a minimum of three days apart (mean days interval between sessions: 6.2, range: 3–14). Each session consisted of four exercises performed on standard weight-stack machines: 1) leg-press (60˚ inclination), 2) chest-press (Technogym, Barcelona, Spain), 3) knee-extension, and 4) lat-pulldown (Life Fitness, Illinois, USA). These exercises were selected as they target the major muscle groups of both the upper and lower body. Exercise order was blocked-randomized so that leg-press was always performed prior to knee-extension. The assigned sequence was consistently performed by each participant and the equipment settings were recorded and maintained throughout the experimental sessions. Exercise execution and form were maintained throughout the sessions. This included consistent hand and feet placement, seat heights, and joint angles, in addition to repetition duration (approximately one second concentric phase and two seconds eccentric phase) in all exercises, which were confirmed by the experimenter across all sessions. Participants were asked to refrain from a strenuous exercise session on the day before sessions. They were also asked to have a fair night sleep and a light meal approximately two hours before sessions. All data were collected by the same two experimenters, at approximately the same time of day (±2 hours), with a consistent room temperature of 22 degrees Celsius.

## Self-report measures

Affective valence was measured via the Feeling Scale (FS), an eleven points bipolar scale ranging from +5 ('very good') through zero ('neutral') to -5 ('very bad') [44]. The experimenter

presented the scale to the participants before and after each set asking the question: "How do you feel?" (also written at the top of the scale). Enjoyment was measured at the end of each session using the Exercise Enjoyment Scale, a seven-points Likert scale ranging from 1 ('not at all') to 7 ('extraordinarily') [45]. The experimenter presented the scale to the participants asking the question: "How much did you enjoy the exercise session?" (also written at the top of the scale). At the end of the third session, the experimenter asked participants the question: "If you had to choose one of the two conditions for your future workouts, which one would you prefer and why?". The experimenter documented participants' answers via a tape-recorder. Participants' responses were transcribed to a data file, translated to English, and edited for coherence by the first author. The same question was introduced to participants again 48 hours later via a text message to allow for short-term effects of the last experimental condition to fade (e.g., arousal, heartrate, etc.). After receiving the responses, we examined if any preference changes occurred. If a participant changed her mind, we coded her last response for analysis and documented any additional qualitative information in a designated column on our data file. We then aggregated the data and extracted the underlying preference themes, in line with Halperin et. al. [46]. All single-item scales went through common validation procedures prior to implementation [8].

## 1RM prediction and familiarization (session 1)

Participants were briefed about the general study design, signed all required forms, were weighed, and familiarized with the self-report scales. They were then instructed to perform a five-minutes warmup walk on a treadmill at a self-selected pace, followed by a general warmup consisting of dynamic stretching and calisthenics exercises. This warmup was identical in all experimental sessions. Then, a 5RM prediction protocol took place based on Brzycki's prediction equation [47]. We used this approach as the Brzycki prediction equation commonly leads to accurate 1RM and since lifting lighter loads can be less intimidating for inexperienced trainees [48, 49]. Participants performed two warmup sets of eight to ten repetitions using a light load which was selected by the experimenter. Participants then performed sets of five repetitions with increasing load until TF was reached. Roughly two minutes of rest were provided between sets. We decided on this rest period to keep the length of this session within a reasonable timeframe, and in view of other studies that have used similar, or shorter rest periods, when implementing 1RM prediction protocols [50]. The maximal load for five repetitions was typed into a web-based calculator (www.ExRx.net) where 1RM prediction values and derivative percentages were calculated. This protocol was repeated for all exercises, in the same order assigned to each participant (see above), with three-minutes of rest between them. To gain familiarity with the FS, participants rated it before and after a few sets.

## Experimental conditions (sessions 2–3)

Participants were first reminded of the single-item scales and performed the general warmup protocol. Thereafter, two specific warmup sets were performed using a light weight selected by the experimenter prior to each exercise (eight to ten repetitions) with approximately one-minute rest between them. Then, three sets were performed using 70% of participants' predicted 1RM, with two-minutes of rest between each set. Three minutes of rest were provided between the different exercises. In the predetermined condition participants were instructed to perform ten repetitions in each set, whereas in the ERF condition, they performed as many repetitions as required until they felt they were two repetitions away from TF. FS scores were collected before and ~5 seconds after each set in both conditions whereas enjoyment scores were

collected after each session. Finally, the open-ended question of preference was presented at the end of the third session in person and 48 hours later via a text message.

## Statistical analyses

Excluding the preference outcome and number of repetitions completed in the predetermined condition, we inspected the normality of the data via kurtosis and skewness inspection, in which skewness < 2 and kurtosis > 7 were considered as substantial deviations from normality [51]. Unless noted otherwise, data is presented as means ± standard deviation (SD). Single item scales were treated as continuous variables following the recommendations of Rhemtulla et al. [52]. We compared the FS scores between the two conditions using the mean absolute scores across sets and exercises, and the mean difference in FS scores (by subtracting the post-set score from the pre-set score for each set in each exercise) using paired t-tests. To further examine the effects of sets, exercises, condition, and their interactions on FS ratings after each set, while holding constant the level of FS ratings before sets-initiation, we tested a mixed regression model of the following form with a random intercept (nested within participants):

$$Post\text{-}set\ FS\ \sim\ condition\ X\ set\ X\ exercise\ +\ Pre\text{-}set\ FS$$

We compared the mean overall enjoyment levels in each condition using paired t-tests. We tested whether the proportion of approach preferences significantly differed from what is expected by chance (0.50) using a binomial test. We compared the mean number of repetitions performed in the ERF condition relative to the fixed ten repetitions in the predetermined condition using a one sample t-test.

Both p-values and 95% confidence intervals (CIs) were calculated and reported for all outcomes. Cohen's d effect sizes were reported when appropriate. For the FS and enjoyment outcomes we calculated Cohen's d as $\dfrac{M_{diff}}{\sqrt{{S^2_{pred}+S^2_{ERF}}\big/2}}$ (the mean differences divided by the average SD across conditions), and for the number of repetitions it was calculated as $t/\sqrt{N}$ [53]. Cohen's d were interpreted using the following criteria: small 0.2–0.5, moderate 0.5–0.8, and large >0.8 [54]. We considered effect sizes smaller than 0.2 as trivial. Binomial and t-tests were carried out with Jamovi (version 1.2.17) and the mixed regression analysis was carried out with R (version 4.0.3) and the lme4 package.

## Results

Twenty participants completed the three experimental sessions. We excluded a datum of knee-extension as it invoked knee pain in one participant and a datum of lat-pulldown of a different participant who performed the exercise on a different machine (with one rather than two pully strings), causing the load to be ~50% lighter than planned. All dependent variables were normally distributed (skewness < 2, kurtosis < 7). The mean FS scores across sets and exercises was slightly higher in the predetermined (3.29±0.89) compared to ERF (3.01±0.95) condition (95%CI [0.09, 0.46], p = 0.006, ES = 0.29). (Fig 1). The mean difference of the pre-post FS scores were comparable in the predetermined (0.27±0.72) and ERF (0.18±0.80) conditions (95%CI [(-0.10,0.30], p = 0.331, ES = 0.13). Table 2 presents the mixed regression results examining the effects of sets, exercises, condition, and their interactions on FS ratings after each set, with the pre-set level of FS rating held constant.

Mean enjoyment scores were slightly higher in the predetermined (5.70±0.80) compared to ERF (5.40±0.94) condition (95%CI [-0.04,0.6], p = 0.081, ES = 0.34). Twelve participants preferred the predetermined condition compared to eight that preferred the ERF condition

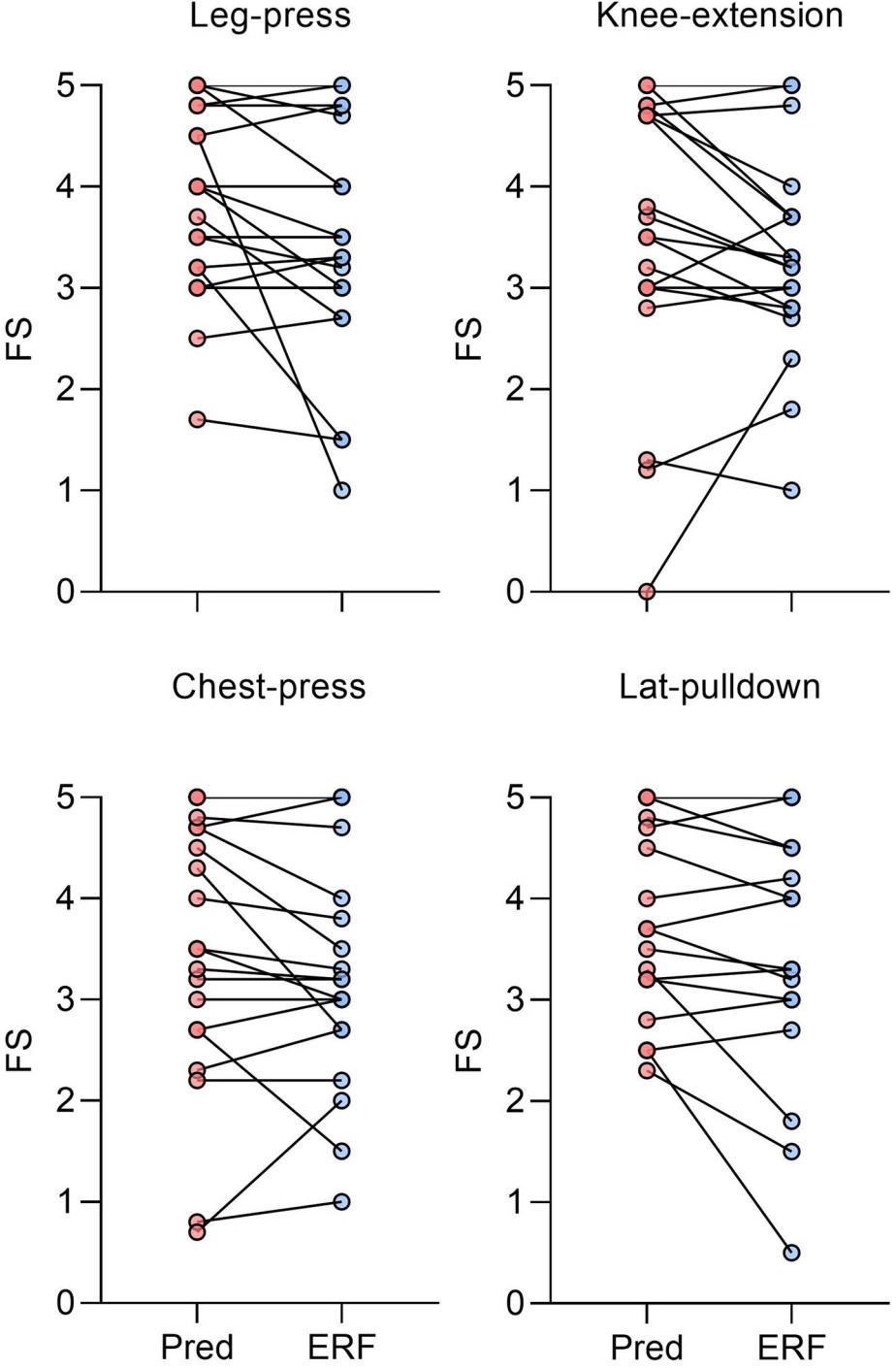

**Fig 1. Mean FS scores (pre and post sets) of each exercise between experimental condition.** Note that n = 20 for the leg-press and chest-press and n = 19 for the knee-extension and the lat-pulldown. *ERF- Estimated Repetitions to Failure; Pred- Predetermined; FS- Feeling Scale.*

(p = 0.261). We observed that participants completed less than the prescribed ten repetitions in 13% of occasions (mostly by 1–2 repetitions). Accordingly, we also compared the number of completed repetitions between conditions using paired, and one sample t-tests. Given that the results of both these tests were similar, we only report the one sample t-test results in

**Table 2. Mixed model regression results.**

| Variable | Estimate (*b*) | SE | t-statistic (df) | p-value | 95% CI |
|---|---|---|---|---|---|
| Condition (ERF vs. Predetermined) | 0.19 | 0.22 | 0.87 (425) | 0.383 | -0.24, 0.64 |
| Exercise (Knee-extension vs. Chest-press) | -0.40 | 0.22 | -1.77 (425) | 0.078 | -0.88, 0.64 |
| Exercise (Lat-pulldown vs. Chest-press) | 0.21 | 0.22 | 0.95 (425) | 0.344 | -0.25, 0.68 |
| Exercise (Leg-press vs. Chest-press) | 0.10 | 0.22 | 0.45 (425) | 0.656 | -0.32, 0.52 |
| Set 2 vs. Set 1 | -0.01 | 0.22 | -0.05 (425) | 0.958 | -0.46, 0.43 |
| Set 3 vs. Set 1 | -0.01 | 0.22 | -0.08 (425) | 0.937 | -0.49, 0.40 |
| Condition X Exercise (knee- extension) | 0.06 | 0.32 | 0.20 (425) | 0.845 | -0.53, 0.71 |
| Condition X Exercise (Lat-pulldown) | -0.38 | 0.32 | -1.20 (425) | 0.230 | -1.01, 0.24 |
| Condition X Exercise (Leg-press) | 0.01 | 0.31 | 0.06 (425) | 0.955 | -0.60, 0.62 |
| Condition X Set 2 | -0.06 | 0.31 | -0.20 (425) | 0.839 | -0.68, 0.53 |
| Condition X Set 3 | -0.06 | 0.31 | -0.20 (425) | 0.839 | -0.65, 0.55 |
| Exercise (knee-extension) X Set 2 | 0.19 | 0.32 | 0.60 (425) | 0.552 | -0.43, 0.82 |
| Exercise (Lat-pulldown) X Set 2 | -0.13 | 0.32 | -0.44 (425) | 0.664 | -0.76, 0.50 |
| Exercise (Leg-press) X Set 2 | 0.13 | 0.31 | 0.41 (425) | 0.683 | -0.50, 0.78 |
| Exercise (Knee- extension) X Set 3 | 0.59 | 0.32 | 1.85 (425) | 0.066 | -0.04, 1.18 |
| Exercise (Lat-pulldown) X Set 3 | 0.16 | 0.31 | 0.51 (425) | 0.607 | -0.44, 0.82 |
| Exercise (Leg-press) X Set 3 | 0.33 | 0.31 | 1.06 (425) | 0.291 | -0.28, 0.93 |
| Condition X Exercise (Knee-extension) X Set 2 | 0.02 | 0.45 | 0.05 (425) | 0.957 | -0.85, 0.88 |
| Condition X Exercise (Lat-pulldown) X Set 2 | 0.25 | 0.45 | 0.55 (425) | 0.583 | -0.65, 1.15 |
| Condition X Exercise (Leg-press) X Set 2 | -0.21 | 0.44 | -0.48 (425) | 0.632 | -1.15, 0.73 |
| Condition X Exercise (Knee-extension) X Set 3 | -0.28 | .45 | -0.63 (425) | .526 | -1.17, 0.62 |
| Condition X Exercise (Lat-pulldown) X Set 3 | -0.18 | .45 | -0.59 (425) | .551 | -0.70, 1.05 |
| Condition X Exercise (Leg-press) X Set 3 | -0.25 | .44 | -0.58 (425) | .564 | -1.13, 0.55 |

SE–standard error CI- Confidence interval

Table 3. The number of repetitions completed by each participant in each exercise under the ERF condition is illustrated in Fig 2. Examples of participants' responses to the open-ended question regarding their preferences are presented in Table 4. Note that only one participant changed her preference from the ERF to the predetermined condition between the end of the session to the text message (48 hours later).

**Table 3. Mean ± SD values for repetitions in the ERF condition compared to the fixed ten repetitions assigned in the predetermined condition.**

| | Repetitions (Mean±SD) | Mean difference (95%CI) | p-value | Effect size |
|---|---|---|---|---|
| Overall | 10.48±2.66 | 0.48 (-0.76, 1.73) | 0.421 | 0.18 |
| Leg-press | 16.90±6.61 | 6.90 (3.80, 9.99) | <0.001 | 1.04 |
| Knee-extension | 8.44±2.13 | -1.56 (-2.59, -0.53) | 0.005 | -0.73 |
| Chest-press | 7.93±1.84 | -2.07 (-3.92, -1.20) | <0.001 | -1.12 |
| Lat-pulldown | 8.70±2.73 | -1.29 (-2.61, 0.02) | 0.053 | -0.47 |

Mean difference, confidence intervals, p-values and Cohen's d effect sizes are reported. Note that n = 20 for the leg-press and chest-press and n = 19 for the knee-extension and the lat-pulldown.

ERF- Estimated Repetitions to Failure

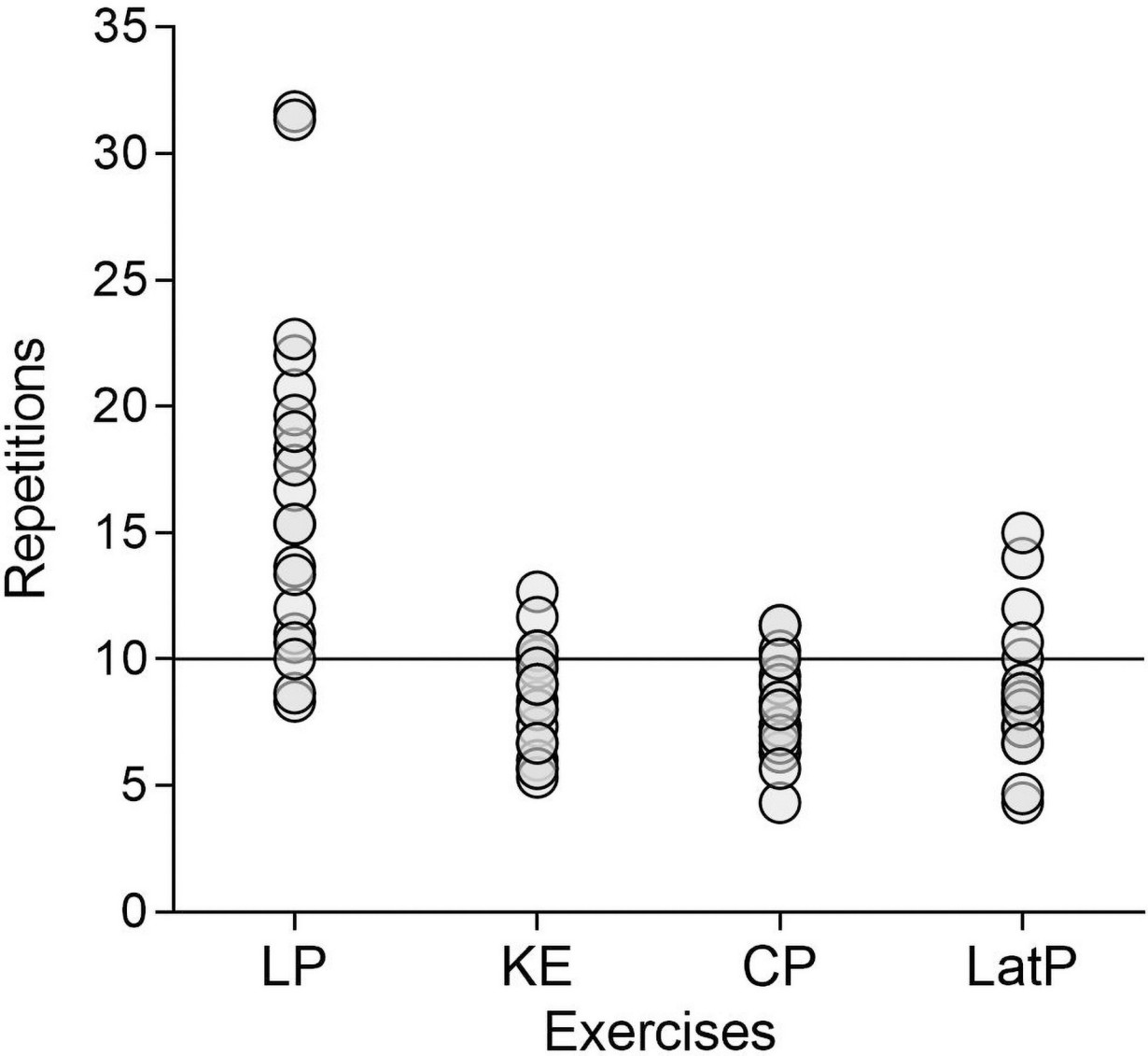

**Fig 2. Repetitions number performed in the ERF condition in relation to the fixed ten repetitions in the predetermined condition (continuous horizontal line).** Circles represent the number of repetitions performed in each exercise by each participant. *ERF: Estimated Repetitions to Failure, LP: Leg-press, KE: Knee extension, CP: Chest press, LatP: Lat pulldown.*

## Discussion

In this study we compared the affective responses during RT using different repetition prescription approaches: the predetermined approach, in which the number of completed repetitions were fixed, and the ERF approach, in which participants were required to terminate the set two repetitions away from TF. The primary outcome were affective responses, collected during and after each session. We collected FS scores before and after each set, enjoyment levels experienced after each session, and participant's approach preferences. The secondary outcome was the number of repetitions completed in each condition. Overall, we observed negligible differences between the conditions in the psychological outcomes, and some difference in the number of repetitions completed in the ERF condition. These findings shed light on the subjective experiences of trainees during RT, and point to future research directions.

**Table 4. Examples of participants' responses to the question: "Which training approach did you prefer?".**

| Predetermined selection | ERF selection |
|---|---|
| • Selecting the number of repetitions was confusing. It made me want to quit sooner. | • It was psychologically challenging for me not to give up. The predetermined number was less challenging. |
| • I like to know where I am going and the endpoint of the set. | • I feel I know how to listen to my body. It feels unpleasant when I am pushed. |
| • When I am tired, I prefer someone telling me what to do. It makes it easier to adhere and this way I am less dependent on my mood. | • I trust I can adequately challenge myself. I know what the most suitable effort is, and the right number of repetitions for me. |
| • I think the instructor knows better than me. It was easier to perform this way. | • The instructor cannot identify my true state like I can during the set. |

Participants' responses were translated from Hebrew and edited for coherence.

In contrast to our expectation, we observed trivial to small differences between conditions in the FS and enjoyment scores, with a small trend favoring the predetermined condition. These results mostly suggest that both approaches led to similar acute affective-valence and enjoyment levels. These trivial to small differences could also stem from the study's duration which lacked the required resolution to capture differences. The similarity in affective responses is also consistent with–and can partly explain–the approximate even split in approach preferences. Two main themes emerged when analyzing participants' answers concerning the approach they preferred: 1) the need to have a clear set endpoint, and 2) the need to make decisions autonomously (Table 4). These two themes are also known to play important roles in motor performance [21, 55] and human motivation [14, 20]. Those who preferred the predetermined condition, might have experienced uncertainty and confusion when allowed to self-regulate the number of repetitions and therefore preferred the predetermined condition. As expressed by one participant: *"This condition is more organized. Figuring out the number of repetitions on my own was confusing and felt inaccurate"*. Those who preferred the ERF condition may have wanted to make decisions during the RT session by regulating their effort. As expressed by another participant: *"It was harder for me to perform an imposed number of repetitions. I prefer to be aware of my abilities and decide when to stop"*. We note that the two reasons leading participants to prefer one approach or the other are not necessarily mutually exclusive. We assume that both can co-exist, yet their relative importance differs between participants.

In the secondary analysis, the mean number of repetitions completed across exercises was comparable. However, when examining the exercises individually (Table 3), in three exercises the number of repetitions was lower in the ERF condition by approximately one repetition as opposed to the leg-press where considerably more repetitions were completed in the ERF condition (17 vs. 10). Despite completing dissimilar number of repetitions across exercises under the ERF condition, participants' perceptions of distance from TF were similar. That is, assuming participants were able to accurately predict TF, then following ERF suggest that they reached comparable actual effort across exercises as indicated by the similar proximity to TF. Conversely, completing ten repetitions across exercises in the predetermined condition suggest that proximity to TF differed between exercises, which led to dissimilar actual effort. Hence, the ERF approach may be advantageous when the goal is to reach comparable degrees of actual effort between exercises.

While a relationship between repetitions number, distance from failure, and FS ratings has been shown to exist [8], when inspecting this relationship in the current study an unclear picture emerges. Whereas the slightly lower FS scores in the ERF condition may be related to the

considerably higher repetitions completed in the leg-press, this pattern was inconsistent. This is because in the remaining three exercises FS scores were slightly lower in the ERF condition despite completing less than ten repetitions on average (see Fig 2). It is possible that this relationship was partly masked by other factors, such as the need for a clear endpoint or the need to be guided (Table 4). These results highlight the need to further inspect how different RT variables interact with one another and influence affective responses.

This study has a number of methodological aspects worthy of discussion. First, we did not verify the actual number of repetitions to TF. This decision was based on the notion that such a requirement may have been overly demanding for this cohort. Moreover, our aim was to ecologically examine the influence of different prescriptions on affective responses rather than inspect prediction accuracy. Therefore, we followed similar RT protocols, in which sets are not taken to failure [22, 39]. Second, our sample consisted of women inexperienced in RT but with experience in Pilates. It is possible that their former background and gender, influenced their estimation, affect, and preferences. This limits our ability to generalize the results of the current study to other populations. Future studies including males and completely untrained participants will shed more light on this issue. Third, we used a prediction protocol to identify 1RM and did not include a familiarization session of this protocol, both of which could have led to some inaccuracies in identifying the true 1RM. The within-subject design we implemented confirmed that all participants lifted the same loads under both experimental conditions although the high number of repetitions in the leg-press could have been a result of not reaching true 5RM in the first session. This might have caused an inaccurate prediction of 1RM and consequently more repetitions performed under the ERF condition. Finally, given that this study was a cross-over design composed of two experiential sessions, future studies should compare the different approaches in a longitudinal manner to develop a deeper understanding of the long-term implications of these prescription approaches.

## Conclusion

We observed that both prescription approaches elicited similar levels of affective-valence, enjoyment, and an approximate even split of approach preferences. Approximately half of the participants preferred the predetermined approach as, according to them, it provided clear and certain endpoints, while the other half preferred the ERF approach as it heightened their sense of autonomy and control. While the mean number of repetitions across all exercises was similar, under the ERF condition participants demonstrated greater variability in repetition-numbers between participants and exercises. Since they maintained a similar proximity to TF, the invested effort across exercises was likely better standardized compared to the predetermined approach. Given these results, RT coaches can attempt to optimize the training experience by introducing both approaches and selecting one or the other based on their trainees' preferences.

## Supporting information

**S1 Data.**
(XLSX)

## Acknowledgments

This study was conducted at the Tel Aviv University Sports Center in partial fulfillment of Hadar Schwartz's PhD degree. The authors would like to express their gratitude to the Tel Aviv University Sports Center management and staff for their assistance.

## Author Contributions

**Conceptualization:** Hadar Schwartz, Aviv Emanuel, Israel Halperin.

**Formal analysis:** Aviv Emanuel.

**Funding acquisition:** Israel Halperin.

**Investigation:** Hadar Schwartz, Aviv Emanuel, Isaac Isur Rozen Samukas.

**Methodology:** Hadar Schwartz, Israel Halperin.

**Project administration:** Hadar Schwartz, Isaac Isur Rozen Samukas, Israel Halperin.

**Supervision:** Isaac Isur Rozen Samukas, Israel Halperin.

**Writing – original draft:** Hadar Schwartz, Aviv Emanuel.

**Writing – review & editing:** Israel Halperin.

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
