## [Decision Letter · Decision Letter 0]

25 May 2021

PONE-D-21-04923

Exploring the acute affective responses to resistance training: a comparison of the predetermined and the estimated repetitions to failure approaches

PLOS ONE

Dear Dr. Halperin,

Thank you for submitting your manuscript to PLOS ONE. After careful consideration, we feel that it has merit but does not fully meet PLOS ONE’s publication criteria as it currently stands. Therefore, we invite you to submit a revised version of the manuscript that addresses the points raised during the review process.

We look forward to receiving your revised manuscript.

Kind regards,

Daniel Boullosa

Academic Editor

PLOS ONE

Journal Requirements:

Reviewers' comments:

Reviewer's Responses to Questions

**Comments to the Author**

1. Is the manuscript technically sound, and do the data support the conclusions?

Reviewer #1: Yes

Reviewer #2: Yes

2. Has the statistical analysis been performed appropriately and rigorously? 

Reviewer #1: Yes

Reviewer #2: Yes

3. Have the authors made all data underlying the findings in their manuscript fully available?

Reviewer #1: Yes

Reviewer #2: Yes

4. Is the manuscript presented in an intelligible fashion and written in standard English?

Reviewer #1: Yes

Reviewer #2: Yes

5. Review Comments to the Author

Reviewer #1: I commend the authors for conducting a well-thought experiment examining the acute affective responses to resistance training. This is an important piece of research that answers several important questions regarding resistance training prescription from a public health perspective, and highlights some avenues for future research. The manuscript is well written, ideas and methods are clearly presented, results appropriately discussed, and limitations acknowledged. However, I do have some comments which could help to strengthen the manuscript further. For more specific comments, please see below.

Abstract

Line 25-26: This parenthetical statements reads like a statement to someone, perhaps revise it to read “(e.g., termination of the sets two repetitions from failure)”, so that it’s more descriptive.

Line 33: Please replace “before” with “away from”.

Introduction

Line 56: Is there a reason why one repetition maximum is written using capital letters?

Line 76: You say “For example, feelings of boredom and monotony in the case of not fulfilling the repetitions…”. Don’t you think that feelings of boredom and monotony are more related to the long-term considerations such as exercise selection, progression schemes etc rather than some acute considerations? Regardless of whether relative effort is taken into account during RT prescription or not, a given RT program can still lead to boredom and monotony. Also, one can fail to fulfil his or her repetition potential even if the relative effort is taken into account with training prescription due to a number of other factors that are beyond our control. So, perhaps this example could be revised to include a reference to a “fixed number of repetitions” paradigm (e.g., constantly doing 3 sets of 10 can lead to …). This becomes even more relevant after reading the second part of the sentence which, to me, suggests that premature TF can happen if relative effort is not taken into account (e.g., people are instructed to do 10 reps, but they only do 8 because that’s all they can do which can then lead to disappointment). I would just revise this to better reflect problems associated with fixed repetitions as it’s obvious that you are trying to build that case in the introduction.

Line 103: I would revise “the topic remains unexamined” because you gave examples where it actually was examined, so it’s a bit contradictory. Perhaps, “…psychological outcomes, this topic remains relatively unexplored (29,32).”

Line 105: Please revise this part of the sentence as follows “… the effects of the ERF and the predetermined RT prescription approaches on acute affective responses…”.

Line 108: Please remove “the” before “different exercises”

Line 123: Please remove “e.g., muscle soreness” since muscle soreness can happen even without reaching failure – in fact, even when further away from it. Muscle soreness is also related to the familiarity – physiologically speaking – with the task, not just “difficulty”. As far as I’m concerned you don’t need an example, but if you want to provide some, I would advise going with fatigue, perception of pain or discomfort.

Lines 190-193: Perhaps, these sentences could be moved to go after describing “a 5RM test”? I believe that would follow a more appropriate sequence of events.

Statistical analysis

I’m unsure how you treated the data from all the sets? You aggregated and then compared the data from all the sets and exercises? You mention in this section that comparisons were made “by subtracting the post-set score from the pre-set score for each set of each exercise”. However, results for each set are not presented. Can you please expand on this in the manuscript?

This above also makes me wonder why you haven’t opted for a factorial design since you already measured variables of interest after every set? I completely understand if this is something that you were not interested apriori. However, even if you were not interested in the sets (apriori – which is, again, completely fine), you can still evaluate the main effect of the condition and include specific comparisons with corrections that are only related to the main effect of condition (perhaps, your main interest). This could be a more robust approach then doing a series of t-tests (regardless of the correction applied). Perhaps, this could be re-analysed and checked whether it makes a difference? Linear-mixed effects modelling is another option here, but since your design seems to be quite balanced, it would probably complicate things without adding much value.

Lines 234-236: I appreciate that you are transparent with regards to missing data, but since you used t-tests, how did you deal with missing data? Did you entirely exclude participants’ data who missed information from their knee extension and lat pulldown sessions, respectively?

Results

I’m wondering, since you already measured all the affective responses after each set of each exercise, why you didn’t report your results broken down by the exercise (and even sets)? This could have unpacked the potential effects of the number of repetitions performed in a given exercise (or set) on the affective responses. Perhaps, doing ~16 repetitions vs ~ 10 repetitions in the leg press exercise affected the psychological outcomes. For instance, if affective valence and enjoyment were not in favour for ERF condition after the leg press exercise, but they were in favour for others, one could argue that the number of repetitions completed confounded the findings. I understand that the manuscript is already packed with the information and complicating it further might not be necessary, but it might still be something worthy of consideration or discussion?

Line 224: You said that the open-ended question of preference was presented at the end of the third session in person and 48 hours later via a text message. Which one did you take for the analysis (or how did you approach data aggregation) and why?

Line 239-240: Please revise this sentence to read: “…we observed that participants completed less than the prescribed 10 repetitions in 13% of occasions (mostly by 1-2 repetitions).”

Table 4: Please check whether this response “The instructor can't identify my true state as the trainee during the set” was correct? Should it maybe say “The instructor can't identify my true state like I can do during the set” or something along these lines?

Discussion

Line 258: Please revise this sentence to read: “…we compared the affective responses during RT using different repetition prescription approaches:…”

Line 261: Please replace “was” with “were”.

Lines 286-287: Perhaps, a concluding statement indicating application of your specific findings here would strengthen the message of the paragraph?

Lines 297-299: I would remove this sentence as we don’t have enough evidence to say “most studies…”. In addition, one of the references you used (number 50) to support your statement stated the following in their practical application: “our findings suggest that RPE accuracy has a direct relationship with training experience; thus, a learning curve likely exists with novice trainees”. In that regard, it seems like we have a conflicting evidence, if nothing, so I would delete this sentence.

Lines 299-305: I believe that discussing your finding “When examining the number of repetitions completed in the predetermined condition across sets, we noted that in 13% of occasions participants completed less than the prescribed 10 (mostly by 1-2 repetitions)” would strengthen some of your arguments here even more.

Lines 306-326: I just want to compliment you here for listing all the limitations of your study (some of which might not even be a limitation given the research question). This level of honesty and consideration is not very common – respect.

Lines 333-334: Please revise this sentence to read: “While the mean number of repetitions across all exercises was similar, participants completed fewer repetitions in some exercises but considerably more in one exercise during the ERF condition”.

Ivan Jukic

Reviewer #2: Thank you for the opportunity to review this submission. This was a wonderfully simple study design addressing an important practical question that is well worth asking. I particularly appreciate the mixed methods approach which is something lacking in our field and a real strength of the work. The authors have already addressed previous concerns of other reviewers. I have some of my own suggestions below that I hope will help to improve the manuscript and feel that if these are addressed the manuscript would be a valuable addition to the literature.

My primary suggestion is to frame the study throughout in an exploratory manner given that an explicit a priori power analysis was not conducted for null hypothesis significance testing. The authors provide a very honest appraisal of their sample size justification which is often absent from most studies in the field where it is certainly the reality. It may be worth reviewing and citing this recent work from Daniel Lakens to support your resource constraint justification (https://psyarxiv.com/9d3yf/). Given this, I would also recommend removing p values from the manuscript and instead focusing on an estimation-based approach and interpretation with respect to uncertainty. I have some suggestions below for analyses and data visualisation in this manner, but in essence I would opt for reporting point and interval estimates and interpreting them cautiously and with respect to what findings may be worth following up on with confirmatory research.

On page 4 lines 76-78 - As the Rhodes and Kates article focuses on affective outcomes, you might want to offer some support that such outcomes are in fact linked to things such as boredom and autonomy in certain contexts (e.g. https://www.ncbi.nlm.nih.gov/pmc/articles/PMC6208645/).

Page 5 lines 97-98 - Prevalence may be even less when 'resistance training' is actually parsed out from other 'muscle strengthening activities' - see discussion in https://bmcpublichealth.biomedcentral.com/articles/10.1186/s12889-017-4209-8. Also, given the female sample it is worth noting that prevalence is typically lower in women which may also give justification for this focus.

Table 1 – I would specify that training sessions refer to ‘non-RT’ training sessions.

Experimental conditions – Could you clarify here what the rest periods used were.

Statistical analyses – Not to say that what has been done here is inherently bad, but I would perhaps opt for a different approach that maximises the use of the data. I have explored the available raw data using this and feel it would probably strengthen the manuscript and also align with its exploratory nature if interpreted cautiously. The primary outcome is FS. For this given you have collected data for multiple sets, and pre and post each set, I would in essence treat this in a similar manner to an RCT with baseline adjustment using ANCOVA, but extended to a within participant design. Also, there are two fixed effects I think worth exploring in interaction with your condition effects. These are the approach-preference categorisation, and also by exercise given the different reps for the leg press. So for feeling scale I would suggest a model of the type:

post_FS ~ (condition * preference * exercise) + pre_FS + (1 | subject_num)

You would in essence have 3 observations (1 per set) for each participant for each exercise and for each condition. From this I would extract the estimated marginal means and their confidence intervals, and then would visualise using a paired estimation plot i.e. plot the paired raw data along with the emmeans and CIs for each condition. Given the exploration of preference and exercise also, I would facet by exercise (a separate panel for each), and color code the data by preference. You can produce model summary tables with fixed and random parameter estimates, p values etc for the supplementary materials if people are interested. But I would focus on the data visualisation and cautious interpretation of the estimates and their uncertainty. You could do the same for enjoyment (but obviously would just have a single data point per participant per condition). The binomial analysis of the preferences is fine as it is. I don’t anticipate any of this will materially change the overall conclusions of the manuscript, but would just better reflect these.

Page 11 line 236 – Can you elaborate on what the technical error was?

Page 15 lines 286-287 – And may also be contextual – see the paper linked above.

Limitations – One of the other reviewers drew issue with the lack of confirmation of ERF accuracy. I would note that you have looked at ecologically valid prescriptions of RT on FS etc. and the aim was not to verify prediction accuracy.

Minor:

Change ‘exercise intensity’ to intensity of effort, and also I would use ‘actual effort’ as opposed to ‘relative effort’ as in places this might be confused with the perception of effort.

Change ‘One Repetition Maximum’ to lower case throughout.

Signed: James Steele

6. PLOS authors have the option to publish the peer review history of their article (what does this mean?). If published, this will include your full peer review and any attached files.

Reviewer #1: **Yes: **Ivan Jukic

Reviewer #2: **Yes: **James Steele

---

## [Author Response · Author response to Decision Letter 0]

9 Jul 2021

Reviewers comments

Reviewer #1: I commend the authors for conducting a well-thought experiment examining the acute affective responses to resistance training. This is an important piece of research that answers several important questions regarding resistance training prescription from a public health perspective, and highlights some avenues for future research. The manuscript is well written, ideas and methods are clearly presented, results appropriately discussed, and limitations acknowledged. However, I do have some comments which could help to strengthen the manuscript further. For more specific comments, please see below.

Response: We thank the reviewer for his kind response and for his valuable comments. We addressed them point by point below.

Abstract

Line 25-26: This parenthetical statements reads like a statement to someone, perhaps revise it to read “(e.g., termination of the sets two repetitions from failure)”, so that it’s more descriptive.

Response: We thank the reviewer and revised the sentence accordingly (line 25). 

Line 33: Please replace “before” with “away from”.

Response: The sentence was revised accordingly (line 32). 

Introduction

Line 56: Is there a reason why one repetition maximum is written using capital letters?

Response: The capital letters were removed (line 54). 

Line 76: You say “For example, feelings of boredom and monotony in the case of not fulfilling the repetitions…”. Don’t you think that feelings of boredom and monotony are more related to the long-term considerations such as exercise selection, progression schemes etc rather than some acute considerations? Regardless of whether relative effort is taken into account during RT prescription or not, a given RT program can still lead to boredom and monotony. Also, one can fail to fulfil his or her repetition potential even if the relative effort is taken into account with training prescription due to a number of other factors that are beyond our control. So, perhaps this example could be revised to include a reference to a “fixed number of repetitions” paradigm (e.g., constantly doing 3 sets of 10 can lead to …). This becomes even more relevant after reading the second part of the sentence which, to me, suggests that premature TF can happen if relative effort is not taken into account (e.g., people are instructed to do 10 reps, but they only do 8 because that’s all they can do which can then lead to disappointment). I would just revise this to better reflect problems associated with fixed repetitions as it’s obvious that you are trying to build that case in the introduction.

Response: We thank the reviewer for this comment. We modified this part of the introduction so it introduces the possible influences of the predetermined approach as they pertain to autonomy support and the intensity of effort. Our aim was to emphasis how the fixed number of repetitions is related to different experiences and the need for individualized prescriptions which will be described in the following paragraph of the manuscript:

“Those differences could alter exercise-adaptations [13] and psychological outcomes such as perceptions of autonomy and competence [14]. For example, trainees might feel unchallenged or bored in case they are prevented from fulfilling the repetitions potential of the set (i.e., by a fixed number of repetitions) [15], or stressed and less competent in the case they are pushed to premature TF (i.e., if they cannot complete the fixed number of repetitions due to natural variability in abilities) [14,16]. These responses, in turn, may also influence the likelihood of adherence to RT programs [17].” (lines 73-79).

Line 103: I would revise “the topic remains unexamined” because you gave examples where it actually was examined, so it’s a bit contradictory. Perhaps, “…psychological outcomes, this topic remains relatively unexplored (29,32).”

Response: We thank the reviewer and revised as suggested (line 105). 

Line 105: Please revise this part of the sentence as follows “… the effects of the ERF and the predetermined RT prescription approaches on acute affective responses…”.

Response: We revised accordingly (line 107). 

Line 108: Please remove “the” before “different exercises”

Response: We revised accordingly (line 109). 

Line 123: Please remove “e.g., muscle soreness” since muscle soreness can happen even without reaching failure – in fact, even when further away from it. Muscle soreness is also related to the familiarity – physiologically speaking – with the task, not just “difficulty”. As far as I’m concerned you don’t need an example, but if you want to provide some, I would advise going with fatigue, perception of pain or discomfort.

Response: We thank the reviewer for this comment and have changed the example to “fatigue and discomfort” (line 125). 

Lines 190-193: Perhaps, these sentences could be moved to go after describing “a 5RM test”? I believe that would follow a more appropriate sequence of events.

Response: We thank the reviewer for this comment. We added a sentence to connect the end of the warmup to the beginning of the 5RM protocol (line 196-197).

Statistical analysis

I’m unsure how you treated the data from all the sets? You aggregated and then compared the data from all the sets and exercises? You mention in this section that comparisons were made “by subtracting the post-set score from the pre-set score for each set of each exercise”. However, results for each set are not presented. Can you please expand on this in the manuscript?

Response: We thank the reviewer for bringing up this point. When analyzing the FS scores across conditions we aggregated the data from all three sets and compared the crude mean difference between conditions in each exercises (Figure 1). However, following both reviewers’ comments we also tested a mixed regression model which examined the effects of sets, exercises, condition, and their interactions on FS ratings after each set. See Table 3 and our following response. We now added this information to the manuscript (lines 229-236). 

This above also makes me wonder why you haven’t opted for a factorial design since you already measured variables of interest after every set? I completely understand if this is something that you were not interested apriori. However, even if you were not interested in the sets (apriori – which is, again, completely fine), you can still evaluate the main effect of the condition and include specific comparisons with corrections that are only related to the main effect of condition (perhaps, your main interest). This could be a more robust approach then doing a series of t-tests (regardless of the correction applied). Perhaps, this could be re-analysed and checked whether it makes a difference? Linear-mixed effects modelling is another option here, but since your design seems to be quite balanced, it would probably complicate things without adding much value.

Response: Due to this and Reviewer 2’s comments, we tested a mixed regression model which examined the effects of sets, exercises, condition, and their interactions on FS ratings after each set (lines 232-236). Regression results are presented in Table 3.

Lines 234-236: I appreciate that you are transparent with regards to missing data, but since you used t-tests, how did you deal with missing data? Did you entirely exclude participants’ data who missed information from their knee extension and lat pulldown sessions, respectively?

Response: When running the t-tests for each exercise we did exclude the missing data. In the knee extension and the lat-pulldown exercises only 19 participants were analyzed. We added this information to the manuscript (Figure 1 legend and Table 3 caption). 

Results

I’m wondering, since you already measured all the affective responses after each set of each exercise, why you didn’t report your results broken down by the exercise (and even sets)? This could have unpacked the potential effects of the number of repetitions performed in a given exercise (or set) on the affective responses. Perhaps, doing ~16 repetitions vs ~ 10 repetitions in the leg press exercise affected the psychological outcomes. For instance, if affective valence and enjoyment were not in favour for ERF condition after the leg press exercise, but they were in favour for others, one could argue that the number of repetitions completed confounded the findings. I understand that the manuscript is already packed with the information and complicating it further might not be necessary, but it might still be something worthy of consideration or discussion?

Response: We thank the reviewer for this comment. First, we created a graphic visualization of the mean FS score for each exercise between the two conditions (Figure 1). We also analyzed our data in line with the reviewers’ recommendations (Table 2). We note that there were no significant effects to any of the factors (i.e., exercise, condition or set). The possible confounding variable of repetition-numbers was also examined and although there was a significant difference for the leg press exercise, this was not a consistent finding or even a trend since in the other three exercises participants performed less than the predetermined ten repetitions with no different influence on affective responses. These data suggest that the trivial to small differences in affect that were found in favor of the predetermined condition were not related to repetition-numbers. We discuss these findings in the discussion section (lines 337-346).

Line 224: You said that the open-ended question of preference was presented at the end of the third session in person and 48 hours later via a text message. Which one did you take for the analysis (or how did you approach data aggregation) and why?

Response: We thank the reviewer for this comment. We sent the question again via text message 48 hours after the last session. First, we verified that the condition preference did not change and then we checked if there was any new qualitative information regarding participants’ preference and added it to our data file in a separate column. Thereafter, we aggregated the data and extracted the main two underlying preference themes presented in Table 4 and in the discussion section (lines 311-324). Overall, there was only one participant who changed her mind. This participant’s preference changed from the ERF approach to the predetermined approach and her explanation was: “when I think about it, I prefer that the instructor will choose the number of repetitions for me, so I can have a clear target”. We now added this information to the manuscript (lines 183-189 and lines 274-275). 

Line 239-240: Please revise this sentence to read: “…we observed that participants completed less than the prescribed 10 repetitions in 13% of occasions (mostly by 1-2 repetitions).”

Response: We revised accordingly (lines 267-268).

Table 4: Please check whether this response “The instructor can't identify my true state as the trainee during the set” was correct? Should it maybe say “The instructor can't identify my true state like I can do during the set” or something along these lines?

Response: We thank the reviewer for this suggestion. We now revised this sentence (Table 4). 

Discussion

Line 258: Please revise this sentence to read: “…we compared the affective responses during RT using different repetition prescription approaches:…”

Response: We revised accordingly (line 294-295). 

Line 261: Please replace “was” with “were”.

Response: We revised accordingly (line 297). 

Lines 286-287: Perhaps, a concluding statement indicating application of your specific findings here would strengthen the message of the paragraph?

Response: We thank the reviewer and added a closing sentence to this paragraph:

“These findings shed light on the subjective experiences of trainees during RT and point to future research directions” (lines 303-304).

Lines 297-299: I would remove this sentence as we don’t have enough evidence to say “most studies…”. In addition, one of the references you used (number 50) to support your statement stated the following in their practical application: “our findings suggest that RPE accuracy has a direct relationship with training experience; thus, a learning curve likely exists with novice trainees”. In that regard, it seems like we have a conflicting evidence, if nothing, so I would delete this sentence.

Response: We thank the reviewer for this comment. We removed this sentence from the manuscript.

Lines 299-305: I believe that discussing your finding “When examining the number of repetitions completed in the predetermined condition across sets, we noted that in 13% of occasions participants completed less than the prescribed 10 (mostly by 1-2 repetitions)” would strengthen some of your arguments here even more.

Response: We thank the reviewer for this comment. Since this paragraph slightly changed, we respectfully decided not to include this sentence in the paragraph. See relevant changes in lines 338-347.

Lines 306-326: I just want to compliment you here for listing all the limitations of your study (some of which might not even be a limitation given the research question). This level of honesty and consideration is not very common – respect.

Response: We thank the reviewer for this comment. 

Lines 333-334: Please revise this sentence to read: “While the mean number of repetitions across all exercises was similar, participants completed fewer repetitions in some exercises but considerably more in one exercise during the ERF condition”.

Response: We revised accordingly (line 373-375). 

Ivan Jukic

Reviewer #2: Thank you for the opportunity to review this submission. This was a wonderfully simple study design addressing an important practical question that is well worth asking. I particularly appreciate the mixed methods approach which is something lacking in our field and a real strength of the work. The authors have already addressed previous concerns of other reviewers. I have some of my own suggestions below that I hope will help to improve the manuscript and feel that if these are addressed the manuscript would be a valuable addition to the literature.

Response: We thank the reviewer for his kind words and for his valuable comments regarding this manuscript. Below are our detailed responses.

My primary suggestion is to frame the study throughout in an exploratory manner given that an explicit a priori power analysis was not conducted for null hypothesis significance testing. The authors provide a very honest appraisal of their sample size justification which is often absent from most studies in the field where it is certainly the reality. It may be worth reviewing and citing this recent work from Daniel Lakens to support your resource constraint justification (https://psyarxiv.com/9d3yf/). Given this, I would also recommend removing p values from the manuscript and instead focusing on an estimation-based approach and interpretation with respect to uncertainty. I have some suggestions below for analyses and data visualisation in this manner, but in essence I would opt for reporting point and interval estimates and interpreting them cautiously and with respect to what findings may be worth following up on with confirmatory research.

Response: We thank the reviewer for this comment. We now added the suggested reference of sample size justification (lines 132-133). We performed a different statistical analysis as suggested by the reviewers where we did not remove p-values but added confidence intervals for each segment as well as effect sizes when appropriate (see tables 2,3). We also aimed to emphasize the exploratory nature of this experiment throughout the manuscript (for example see lines 106, 112, 349-351).

On page 4 lines 76-78 - As the Rhodes and Kates article focuses on affective outcomes, you might want to offer some support that such outcomes are in fact linked to things such as boredom and autonomy in certain contexts (e.g. https://www.ncbi.nlm.nih.gov/pmc/articles/PMC6208645/).

Response: We thank the reviewer for this excellent reference. We slightly edited the paragraph and cited it in the relevant place:

“Those differences could alter exercise-adaptations [13] and psychological outcomes such as perceptions of autonomy and competence [14]. For example, trainees might feel unchallenged or bored in case they are prevented from fulfilling the repetitions potential of the set (i.e., by a fixed number of repetitions) [15], or stressed and less competent in the case they are pushed to premature TF (i.e., if they cannot complete the fixed number of repetitions due to natural variability in abilities) [14,16]. These responses, in turn, may also influence the likelihood of adherence to RT programs [17].” (lines 73-79)

Page 5 lines 97-98 - Prevalence may be even less when 'resistance training' is actually parsed out from other 'muscle strengthening activities' - see discussion in https://bmcpublichealth.biomedcentral.com/articles/10.1186/s12889-017-4209-8. Also, given the female sample it is worth noting that prevalence is typically lower in women which may also give justification for this focus.

Response: We thank the reviewer for this comment. We inserted this reference to support our statement (line 99).

Table 1 – I would specify that training sessions refer to ‘non-RT’ training sessions.

Response: We revised accordingly (Table 1). 

Experimental conditions – Could you clarify here what the rest periods used were.

Response: Rest periods for the warmup sets were approximately one minute and two minutes between experimental sets. Three minutes rest were provided between exercises. We added this information to the manuscript (lines 213-216). 

Statistical analyses – Not to say that what has been done here is inherently bad, but I would perhaps opt for a different approach that maximises the use of the data. I have explored the available raw data using this and feel it would probably strengthen the manuscript and also align with its exploratory nature if interpreted cautiously. The primary outcome is FS. For this given you have collected data for multiple sets, and pre and post each set, I would in essence treat this in a similar manner to an RCT with baseline adjustment using ANCOVA, but extended to a within participant design. Also, there are two fixed effects I think worth exploring in interaction with your condition effects. These are the approach-preference categorisation, and also by exercise given the different reps for the leg press. So for feeling scale I would suggest a model of the type:

post_FS ~ (condition * preference * exercise) + pre_FS + (1 | subject_num)

You would in essence have 3 observations (1 per set) for each participant for each exercise and for each condition. From this I would extract the estimated marginal means and their confidence intervals, and then would visualise using a paired estimation plot i.e. plot the paired raw data along with the emmeans and CIs for each condition. Given the exploration of preference and exercise also, I would facet by exercise (a separate panel for each), and color code the data by preference. You can produce model summary tables with fixed and random parameter estimates, p values etc for the supplementary materials if people are interested. But I would focus on the data visualisation and cautious interpretation of the estimates and their uncertainty. You could do the same for enjoyment (but obviously would just have a single data point per participant per condition). The binomial analysis of the preferences is fine as it is. I don’t anticipate any of this will materially change the overall conclusions of the manuscript, but would just better reflect these.

Response: Due to this and Reviewer 1’s comments, we tested a mixed regression model of the form: Post-set FS ~ condition X set X exercise + pre-set FS + (1|subject_num). Regression results including 95% Cis, are presented in Table 2. We chose to replace the preference predictor with the set predictor in line with Reviewer 1’s comment, because we conducted preference measurement after the two conditions were completed. Thus, we thought it was more sensible to assume that affective state affected preferences but not vice-versa. 

Page 11 line 236 – Can you elaborate on what the technical error was?

Response: The technical error occurred as a result of a human-error. The participant performed the exercise on a different machine (with one rather than two pully strings), causing the load to be ~50% lighter than planned. We added this information to the manuscript (lines 254-255). 

Page 15 lines 286-287 – And may also be contextual – see the paper linked above.

Response: We thank the reviewer for this comment. Given that we added a new paragraph to the already relatively long discussion, we think that adding any additional information may take away from the key points. 

Limitations – One of the other reviewers drew issue with the lack of confirmation of ERF accuracy. I would note that you have looked at ecologically valid prescriptions of RT on FS etc. and the aim was not to verify prediction accuracy.

Response: We thank the reviewer for this comment. We added this to the limitations section (lines 350-352). 

Minor:

Change ‘exercise intensity’ to intensity of effort, and also I would use ‘actual effort’ as opposed to ‘relative effort’ as in places this might be confused with the perception of effort.

Response: We thank the reviewer for this comment. We changed exercise intensity to intensity of effort (line 87), and “relative” to “actual” effort (lines 69, 71, 332, 335, 336).

Change ‘One Repetition Maximum’ to lower case throughout.

Response: We revised accordingly. 

Signed: James Steele

---

## [Decision Letter · Decision Letter 1]

3 Aug 2021

Exploring the acute affective responses to resistance training: a comparison of the predetermined and the estimated repetitions to failure approaches

PONE-D-21-04923R1

Dear Dr. Halperin,

We’re pleased to inform you that your manuscript has been judged scientifically suitable for publication and will be formally accepted for publication once it meets all outstanding technical requirements.

Kind regards,

Daniel Boullosa

Academic Editor

PLOS ONE

Additional Editor Comments (optional):

Reviewers' comments:

Reviewer's Responses to Questions

**Comments to the Author**

1. If the authors have adequately addressed your comments raised in a previous round of review and you feel that this manuscript is now acceptable for publication, you may indicate that here to bypass the “Comments to the Author” section, enter your conflict of interest statement in the “Confidential to Editor” section, and submit your "Accept" recommendation.

Reviewer #1: All comments have been addressed

Reviewer #2: All comments have been addressed

2. Is the manuscript technically sound, and do the data support the conclusions?

Reviewer #1: Yes

Reviewer #2: Yes

3. Has the statistical analysis been performed appropriately and rigorously? 

Reviewer #1: Yes

Reviewer #2: Yes

4. Have the authors made all data underlying the findings in their manuscript fully available?

Reviewer #1: Yes

Reviewer #2: Yes

5. Is the manuscript presented in an intelligible fashion and written in standard English?

Reviewer #1: Yes

Reviewer #2: Yes

6. Review Comments to the Author

Reviewer #1: The authors are to be applauded for making significant changes to the manuscript. My comments have been adequately addressed, and the manuscript now makes a clear and significant contribution to the literature.

Ivan Jukic

Reviewer #2: Thank you for your responses and edits to both my comments and those of the other reviewer. I'm happy to endorse the submission in it's current form.

7. PLOS authors have the option to publish the peer review history of their article (what does this mean?). If published, this will include your full peer review and any attached files.

Reviewer #1: **Yes: **Ivan Jukic

Reviewer #2: **Yes: **James Steele

---

## [Editor Report · Acceptance letter]

9 Aug 2021

PONE-D-21-04923R1 

Exploring the acute affective responses to resistance training: a comparison of the predetermined and the estimated repetitions to failure approaches 

Dear Dr. Halperin:

I'm pleased to inform you that your manuscript has been deemed suitable for publication in PLOS ONE. Congratulations! Your manuscript is now with our production department. 

Kind regards, 

on behalf of

Dr. Daniel Boullosa 

Academic Editor

PLOS ONE